# Initial Experience of Cytoreductive Surgery (CRS) and Hyperthermic Intraperitoneal Chemotherapy (HIPEC) in Baltic Country Center

**DOI:** 10.3390/jcm11195554

**Published:** 2022-09-22

**Authors:** Rokas Račkauskas, Augustinas Baušys, Jonas Jurgaitis, Marius Paškonis, Kęstutis Strupas

**Affiliations:** Clinic of Gastroenterology, Nephrourology, and Surgery, Institute of Clinical Medicine, Faculty of Medicine, Vilnius University, Ciurlionio Str. 21, Vilnius 03101, Lithuania

**Keywords:** peritoneal surface malignancies, cytoreductive surgery, hyperthermic intraperitoneal chemotherapy, Peritoneal mesothelioma, colorectal peritoneal metastases, peritoneal carcinomatosis, peritoneum cancer

## Abstract

**Background**: Peritoneal surface malignancies (PSMs) are a heterogenous group of primary and metastatic cancers affecting the peritoneum. They are associated with poor long-term outcomes. Many centers around the world adopt cytoreductive surgery (CRS) and hyperthermic intraperitoneal chemotherapy (HIPEC) in routine clinical practice for these otherwise condemned patients despite a lack of high-level evidence from randomized control trials. This study aimed to investigate and present our 10-year experience with this controversial method, CRS and HIPEC, for PSM in a single tertiary center in a Baltic country. **Methods:** Patients who underwent CRS and HIPEC at Vilnius University Hospital Santaros Klinikos between 2011 and 2021 were included in this retrospective study. Overall survival was the primary study outcome. Secondary outcomes included postoperative morbidity and mortality, and local or systemic recurrence rates. **Results:** Sixty-nine patients who underwent CRS and HIPEC were included in the study. Most patients underwent treatment for peritoneal metastases from colorectal, ovarian, and appendiceal cancers. Six (8.7%) patients received CRS and HIPEC for primary peritoneal neoplasm—pseudomyxoma peritonei. The mean peritoneal carcinomatosis index score was 12 ± 7. Complete cytoreduction was achieved in 62 (89.9%) patients. The mean OS was 39 ± 29 months. The mean survival of patients with PSMs of different origin was as follows: 39 ± 25 (95% CI: 28–50) months for colorectal cancer, 44 ± 31 (95% CI: 30–58) months for ovarian cancer, 32 ± 21 (95% CI: 21–43) months for appendiceal cancer, 422 ± 1 (95% CI: 12–97) months for pseudomyxoma peritonei, and 7 months for gastric cancer. **Conclusions:** The current study demonstrated the results of the CRS and HIPEC program in a single Baltic country tertiary center. Patients who underwent CRS and HIPEC for PSMs achieved moderate survival rates with acceptable postoperative morbidity and mortality risk.

## 1. Background

Peritoneal mesothelioma, pseudomyxoma peritonei, and peritoneal metastases of various cancers are among the main tumors that make up the heterogeneous group of peritoneal surface malignancies (PSMs). The most typical sources of metastasis are colorectal, stomach, and ovarian malignancies [1]. Irrespective of the origin, PSMs are associated with poor long-term outcomes. Historically, the main treatment options for PSM patients were palliative systemic chemotherapy or palliative care [2,3]. However, in recent decades, the notion of futile PSM treatment has evolved. The oncological community has grown interested in a more aggressive and potentially curative treatment approach using cytoreductive surgery (CRS) and hyperthermic intraperitoneal chemotherapy (HIPEC) [4]. Many centers around the world adopt CRS and HIPEC in routine clinical practice for these otherwise condemned patients despite a lack of high-level evidence from randomized control trials (RCTs). Accumulating evidence from observational studies promoted the spread of CRS and HIPEC for PSMs, and now, the treatment is considered the standard of care for primary peritoneal malignancies despite the absence of level 1 evidence due to lack of an effective alternative treatment [5]. However, certain recent high-quality RCTs, such as PRODIGE7, COLOPEC, CYTO-CHIP, and PROFILOCHIP presented conflicting data and questioned whether HIPEC regimens currently in use are helpful for peritoneal metastases [6,7,8,9]. Thus, this study aimed to investigate and present our 10-year experience with this controversial method, CRS and HIPEC, for PSMs in a single tertiary center in a Baltic country.

## 2. Materials and Methods

### 2.1. Ethics

Ethical approval from Vilnius regional biomedical research ethics committee (No. 2020/11-1279-761) was obtained before the start of the study. The waiver for informed consent was given by the authority. The study was conducted according to the Declaration of Helsinki.

### 2.2. Patients and Data Collection

All consecutive patients who underwent CRS and HIPEC at Vilnius University Hospital Santaros Klinikos between 2011 and 2021 were included in this retrospective study. Every patient with a PSM was discussed at multidisciplinary team meetings and decision to perform CRS and HIPEC for PSMs was individualized depending on patients’ general condition (ECOG 0-1), etiology, and the dissemination of the disease. Data on patient characteristics were extracted from the prospectively collected institutional electronic database. They included clinicopathologic characteristics (age; gender; history of previous cancer treatment; origin, number, and size of metastatic lesions; and peritoneal carcinomatosis index (PCI) score) and treatment-related characteristics (length of surgery; blood loss; HIPEC regime; postoperative complications as per Clavien–Dindo classification).

### 2.3. Study Outcomes

The primary outcome of the study was overall survival (OS) in patients with peritoneal carcinomatosis treated with CRS and HIPEC. OS was defined as the time from surgical intervention to death. The secondary outcomes included postoperative morbidity, mortality, and local or systemic recurrence. Data on survival and date of death were collected from the Lithuanian National Cancer Registry.

### 2.4. CRS and HIPEC Regimens

All procedures were performed under general anesthesia. At first, laparotomy was performed, and the extent of CRS and organ resections depended on the dissemination of the disease. After CRS, HIPEC was performed. Different protocols were used for PSMs of different origin:For ovarian and gastric cancer, the HIPEC protocol consisted of Cisplatin at 75 mg/m^2^ with 4 L of 0.9% saline at 42 °C for 60 min;For colorectal cancer, the HIPEC protocol consisted of Oxaliplatin at 460/m^2^ with 4 L of 5% glucose at 42 °C for 45 min until 2017; later, it was changed to Mytomicin C at 30 mg + 10 mg, with 5 L of 1.5% dextrose at 42 °C for 90 min;For primary PSMs (pseudomyxoma peritonei, appendiceal cancer), the HIPEC protocol consisted of Mytomicin C 30 mg + 10 mg, with 5-L 1.5% dextrose at 42 °C for 90 min.

After surgery, all patients were moved to the intensive care unit for further treatment. Typically, if patients vital functions were stable, they were moved to further treatment in a surgical ward on postoperative day 1 or 2.

### 2.5. Statistical Analysis

All statistical analyses were conducted using the statistical program SPSS 24.0 (SPSS, Chicago, IL, USA). Continuous variables are presented as means with standard deviation and an interquartile range. Categorical variables are shown as proportions. Continuous variables were compared using a Mann–Whitney U-test and categorical variables with Pearson’s chi-square or Fisher’s exact test, as appropriate. Overall survival rates were using the Kaplan–Meier method and compared with the log-rank test. Statistical significance was considered when a *p*-value < 0.05 was achieved.

## 3. Results

### 3.1. Baseline Characteristics

In total, 69 patients who underwent CRS and HIPEC were included in the study. The baseline characteristics of patients are shown in Table 1. All patients (100%) had an ASA score of 2 or 3. Most patients underwent treatment for peritoneal metastases from colorectal, ovarian, and appendiceal cancers. Six (8.7%) patients received CRS and HIPEC for primary peritoneal neoplasm—pseudomyxoma peritonei. The mean peritoneal carcinomatosis index (PCI) score at the time of surgery was 12 ± 7. Complete cytoreduction was achieved in 62 (89.9%) patients. After CRS and HIPEC, all patients (100%) received systemic therapy (chemotherapy/biological therapy) prior and after the procedure

### 3.2. Postoperative Morbidity

Postoperative complications occurred in 33 (47.8%) patients, including 11 patients (15.9%) who suffered severe complications (≥Clavien–Dindo 3). The postoperative mortality rate was 1.4% (one patient). A detailed list of complications and their severity according to the Clavien–Dindo classification is shown in Table 2.

### 3.3. Long-Term Outcomes

The mean OS of patients treated with CRS and HIPEC was 39 ± 29 months; however, it greatly depended on the etiology of the disease. The mean survival of patients with PSMs of different origin was as follows: 39 ± 25 (95% CI: 28–50) months for colorectal cancer, 44 ± 31 (95% CI: 30–58) months for ovarian cancer, 32 ± 21 (95% CI: 21–43) months for appendiceal cancer, 42 ± 21 (95% CI: 12–97) months for pseudomyxoma peritonei, and 7 months for gastric cancer. However, these differences failed for significance (Figure 1). The mean time for recurrence was 15 ± 12 months, with no differences among PSMs of different origin (data not shown).

### 3.4. CRS and HIPEC Development in the Study Center

Since the implementation of CRS and HIPEC into clinical practice at our center, there have been ongoing worldwide debates about the effectiveness and benefits of this treatment modality. However, through the study period, with accumulating high-level evidence questioning the effectiveness of CRS and HIPEC in many cancers, the number of procedures tended to drop at our center (Figure 2). Currently, CRS and HIPEC are performed at our center only for primary peritoneal malignancy and appendiceal LAMN and HAMN tumors.

## 4. Discussion

The present study demonstrated the results of CRS and HIPEC for PSMs at a single tertiary center in a Baltic country. The postoperative outcomes achieved in our study in terms of postoperative morbidity and long-term outcomes are comparable to international standards. The present study showed that CRS and HIPEC are safe and feasible with a low mortality rate in medium- or low-volume centers in small-population countries. However, current standard indications for CRS and HIPEC are narrow; thus, clinical trials may be an option to offer this treatment to patients and maintain sufficient center volume.

Since the introduction of CRS and HIPEC for PSMs, it has been postulated to increase the survival of patients with advanced peritoneal surface disease [10,11]. However, recent RCTs showed controversial results, and the rationale for HIPEC in some types of cancer became questionable. The third most prevalent disease in the world is colorectal cancer, and up to 15% of patients have peritoneal metastases [12]. The present study showed that CRS and HIPEC in such patients can achieve a mean survival of 39 months. Such results seem promising, similar to initial reports that showed that CRS and HIPEC offer survival benefits over systemic chemotherapy [13]. However, a recent PRODIGE-7 study showed that HIPEC did not improve survival in colorectal cancer patients who achieved CC0 resection compared with CRS alone but increased morbidity 60 days after surgery. Thus, our satisfactory results achieved using CRS and HIPEC do not encourage the further adoption of the method for colorectal cancer patients [6,14].

Another entity that may be considered for CRS and HIPEC is ovarian cancer. In our study, these patients achieved a mean survival of 44 months. Globally, there are still ongoing debates regarding HIPEC’s role after CRS for primary and recurrent ovarian cancer [15,16]. Some studies showed that patients with stage III epithelial ovarian cancer may benefit from HIPEC after CRS in terms of increased OS and recurrence-free survival [17]. However, the lack of robust evidence resulted in a decreased number of these procedures for ovarian cancer in our center [18].

Primary peritoneal malignancies, such as pseudomyxoma peritonei or malignant mesothelioma, and mucinous appendiceal tumors are among those where CRS with HIPEC is the standard treatment option [19]. Several studies comparing CRS with HIPEC to surgery and debulking alone failed to show HIPEC benefits for long-term outcomes [20,21], but the lack of an effective alternative treatment still precludes abandoning HIPEC for these patients. Thus, the recent clinical practice guidelines recommend that patients with these rare malignancies should be referred to a specialized center for a personalized treatment approach [22]. In our center’s experience, CRS and HIPEC are feasible, safe, and promising for these patients, especially for patients with pseudomyxoma peritonei, as the 10-year OS of these patients was 100%.

Regarding gastric cancer peritoneal carcinomatosis treatment, there is no high-level evidence that would strongly support the use of CRS and HIPEC. Several studies from Asian countries suggested that prophylactic HIPEC in high-risk patients may improve long-term outcomes, but since they are confined to exclusively the Asian population or a small number of included patients, such treatment cannot be considered outside of clinical trials [22,23,24,25].

Taking together the current evidence, CRS with HIPEC is the cornerstone treatment option only for primary peritoneal malignancies (pseudomyxoma peritonei or malignant mesothelioma) and mucinous appendiceal tumors. Other indications, such as peritoneal metastases arising from colorectal, ovarian, and gastric cancers remain controversial. The optimization of HIPEC protocols, new drugs and techniques (i.e., water lavage), and the development and improvement of perioperative care techniques that would reduce postoperative morbidity may expand the indications for CRS with HIPEC, but further studies are needed [26,27,28,29,30]

The main and most important limitation of present study was its retrospective nature which greatly limited the level of evidence. This emphasizes the need for the centralization of such patients with peritoneal surface pathology, thus allowing more evidence to emerge in randomized clinical trial settings.

## 5. Conclusions

The current study demonstrated the results of the CRS and HIPEC program in a single Baltic country tertiary center. Patients who underwent CRS and HIPEC for PSMs achieved moderate survival rates with acceptable postoperative morbidity and mortality risk. CRS with HIPEC is a standard treatment option for primary peritoneal malignancies, and our study confirmed excellent outcomes in these patients. However, the lack of high-level evidence on CRS and HIPEC for most of the peritoneal metastases precludes the utilization of this treatment outside the clinical trial setting.

## Figures and Tables

**Figure 1 jcm-11-05554-f001:**
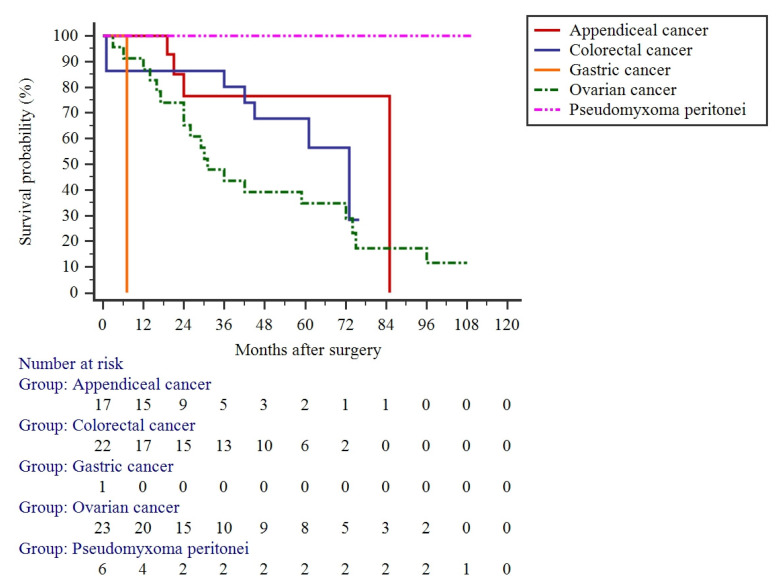
Overall survival of patients who underwent cytoreductive surgery and hyperthermic intraperitoneal chemotherapy for peritoneal surface malignancies.

**Figure 2 jcm-11-05554-f002:**
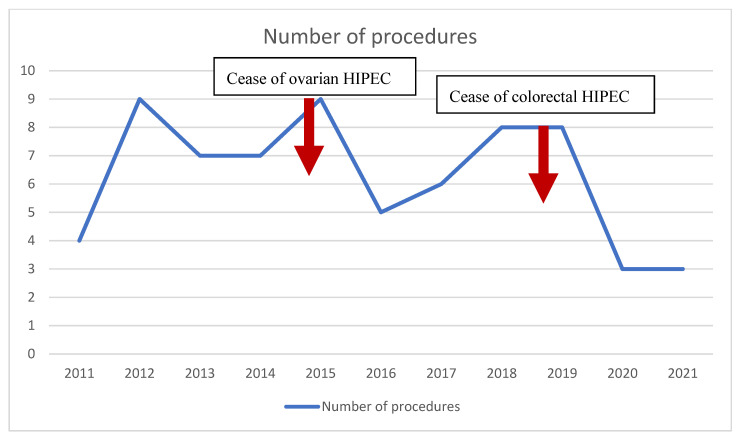
The annual number of cytoreductive surgery and hyperthermic intraperitoneal chemotherapy procedures throughout the study period.

**Table 1 jcm-11-05554-t001:** Baseline clinicopathologic characteristics of patients who underwent cytoreductive surgery and hyperthermic intraperitoneal chemotherapy for peritoneal surface malignancies.

Characteristics of HIPEC Patients		
Sex	Female (n; %)	56 (81.2%)
	Male (n; %)	13 (18.8%)
Mean age ± SD (Q1; Q3), years		57.2 ± 10.89 (49; 67)
Mean hospitalization ± SD (Q1; Q3), days		20.19 ± 14.69 (12; 23.5)
Peritoneal histology (n; %):		
Colorectal		22 (31.9%)
Ovarian		23 (33.3%)
Appendiceal:		17 (24.6%)
- LAMN		13 (18.8%)
- HAMN		4 (5.8%)
Pseudomyxoma peritonei		6 (8.7%)
Gastric		1 (1.4%)
Mean PCI score ± SD (Q1; Q3)		12.2 ± 7.6 (6.25; 15.75)
Mean operation time ± SD (Q1; Q3), min		447.5 ± 152.7 (310; 540)
Mean blood loss ± SD (Q1; Q3), mL		350.7 ± 284.3 (200.0; 500.0)
Cytoreduction completeness:		
- CC0 (n; %) - CC1 (n; %) - CC2 (n; %)		- 62 (89.9%) - 6 (8.7%) - 1 (1.4%)
Complications C–D:		
- No complications - <3 - 3a - 3b - 5		36 (52.2%) 21 (30.4%) 5 (7.2%) 6 (8.7%) 1 (1.4%)

**Table 2 jcm-11-05554-t002:** Postoperative complications after cytoreductive surgery and hyperthermic intraperitoneal chemotherapy for peritoneal surface malignancies.

Complication	Number of Patients	C–D
Atrial fibrillation	2	1
Wound infection	6	2
Urinary tract infection	3	2
Renal insufficiency	3	1
Pancytopenia	2	2
Hydrothorax	5	2
Pneumonia	3	2
Intraabdominal abscess	4	3a
Intraabdominal bleeding	1	3b
Pancreatic fistula	2	3a
Ileus	3	1
Anastomotic leakage	5	3b
Compartment syndrome	1	3b
Postoperative myocardial infarction	1	5

## Data Availability

The datasets used and/or analyzed during the current study are available from the corresponding author upon reasonable request.

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
