# Peer review of "Initial Experience of Cytoreductive Surgery (CRS) and Hyperthermic Intraperitoneal Chemotherapy (HIPEC) in Baltic Country Center"

_jcm, 2022, doi:10.3390/jcm11195554_

Round 1

Reviewer 1 Report

The study showed the effectiveness of  CRS and HIPEC for PSM achieved  relative good survival rates with acceptable postoperative morbidity and mortality risk. It is reasonable. 

Comments:

1) the report  represented the data of Baltic country, and the selection criteria is relative strict. Almost  90% reached CC0. The authors should offered the result of higher and lower PCI and the effectiveness of CC0. 

2) It is relatively short of data, the reading is limited to the tumor, staging, PCI, complication and outcome. IS complicated related to outcome? 

Author Response

Reviewer 1

The study showed the effectiveness of CRS and HIPEC for PSM achieved  relative good survival rates with acceptable postoperative morbidity and mortality risk. It is reasonable.

Comments:

1) the report represented the data of Baltic country, and the selection criteria is relative strict. Almost  90% reached CC0. The authors should offered the result of higher and lower PCI and the effectiveness of CC0.

2) It is relatively short of data, the reading is limited to the tumor, staging, PCI, complication and outcome. IS complicated related to outcome?

At first, we would like to sincerely thank the reviewer for his comments on our paper. We agree with the reviewer, that strict selection of candidates for CRS+HIPEC lead to high rates of CC0. Although, we believe, that such a strict selection of patients is necessary, as this procedure is related to high postoperative morbidity and only selected patients may benefit from it.

In our analysis we dis not observe association between postoperative complications and impaired outcomes, thus did not present these results in the manuscript.

Reviewer 2 Report

This is a very interesting paper, clearly presenting data from a well designed study on an clinically significant topic. There are a few minor problems with English expression and presentation points that need to be addressed. 

1. Line 102, please explain what the PCI score is.

2. Line 125, and Line 197  However instead of Although

3. Line 153, "emerged the knowledge", please rephrase or delete

4. Line 175, Thus instead of Thys

5. Lines 185 and 186. Please rephrase and refer to your study and other studies separately.

Author Response

We would like to thank the reviewer for such a positive comment on our study. As suggested we made necessary corrections for English expression and presentation points.

Reviewer 3 Report

1.-Little casuistry. 2.- Lack of protocols according to tumor origin. 3.- No contribution to the diagnosis and treatment of peritoneal oncological disease. 4.- Relevant bibliography is missing.

Author Response

We would like to express our sincere gratitude for the reviewer for his/her relevant comments on our manuscript. As suggested, we added HIPEC protocols according to tumor origin and updated the bibliography.

Reviewer 4 Report

This study provides a real world description of outcomes of CRS/HIPEC in a region not typically reported. While the paper may be valuable to readers, there are many details that need to be included.

What drugs were used for the HIPEC? How long was the chemoperfusion? What was used to rinse out the chemoperfusate (saline/water)? At what temperature was this performed? What organs were typically removed? How were patients managed peri-operatively/intra-operatively? What co-morbidities did the patient population have, or at least report some sort of comorbid/ASA score so complications can be appropriately interpreted. What was average length of stay? Did patients require the ICU? There are many nuances to the procedure that are not included and should be clarified. Other relevant citations on items pertaining to this include: PMID: 30278972 and PMID: 28622839 to name a few.

2. 

Author Response

All authors of the manuscript would like to thank the reviewer for his/her valuable comments on our paper. As suggested, we explained CRS and HIPEC protocols for different cancers in the “Methods” section. Also, we clarified that all patients were ASA 2-3 score. The average length of stay was 20±14 days (Table 1). In the methods we clarified that all patients went ICU care after CRS+HIPEC according to protocols of institution. Necessary citations were added. We believe that comments of the reviewer helped us to substantially improve the quality of the manuscript.

Round 2

Reviewer 2 Report

The manuscript is acceptable for publication.

Reviewer 3 Report

The scarce casuistry does not allow relevant conclusions to be drawn in relation to the diagnosis and treatment of malignant peitoneal disease. The work is not relevant to be published in this journal.

Reviewer 4 Report

The authors have made substantial changes to the methods and results. Acceptable for publication at this time.